# Prevalence of Swine Gastrointestinal Parasites in Two Free-Range Farms from Nord-West Region of Romania

**DOI:** 10.3390/pathogens11090954

**Published:** 2022-08-23

**Authors:** Mihai-Horia Băieş, Zsolt Boros, Călin Mircea Gherman, Marina Spînu, Attila Mathe, Stefan Pataky, Menelaos Lefkaditis, Vasile Cozma

**Affiliations:** 1Department of Parasitology and Parasitic Diseases, Faculty of Veterinary Medicine, University of Agricultural Sciences and Veterinary Medicine of Cluj-Napoca, Mănăştur Street 3-5, 400372 Cluj-Napoca, Romania; 2Department of Infectious Diseases, Faculty of Veterinary Medicine, University of Agricultural Sciences and Veterinary Medicine of Cluj-Napoca, Mănăştur Street 3-5, 400372 Cluj-Napoca, Romania; 3Agricultural Research and Development Station of Turda, Agriculturii Street 27, 401100 Turda, Romania; 4Laboratory of Microbiology and Parasitology, Department of Veterinary Medicine, School of Health Sciences, University of Thessaly, 43100 Karditsa, Greece; 5Academy of Agricultural and Forestry Sciences Gheorghe Ionescu-Siseşti (A.S.A.S.), Mărăști Boulevard 61, 011464 Bucharest, Romania

**Keywords:** epidemiology, free-range farms, gastrointestinal parasites, swine

## Abstract

Parasitic diseases cause significant economic losses in swine, including free-range swine farms, the number of which in Romania has increased in the last decades. The current study aimed to identify the parasitic profile of swine raised on two free-range (low-input) farms from Transylvania. Nine hundred sixty samples collected from weaners, fatteners, and sows were investigated by flotation, centrifugal sedimentation, modified Ziehl-Neelsen stained fecal smear, modified Blagg technique, and oocyst/egg cultures. The number of oocysts (OPG), cysts (CPG), and eggs (EPG) were counted per gram of fecal matter. The examination revealed parasitic infections with *Balantidium coli*, *Eimeria* spp., *Ascaris suum*, *Trichuris suis*, *Oesophagostomum* spp., *Strongyloides ransomi* and *Cryptosporidium* spp. Prevalence (P) and the mean intensity (MI) of the infections varied according to age, swine category, farm, and season. The overall prevalence in both free-range farms according to the age category was 63.2%—*Eimeria* spp., 70.31%—*B. coli*, 9.38%—*Oesophagostomum* spp., 3.75% *S. ransomi*, and 18.12% *Cryptosporidium* spp. in weaners. In fatteners *Eimeria* spp. revealed a prevalence of 50.93%, *B. coli*—72.5 %, *A. suum*—63.13%, *T. suis*—39.06%, and in sows *Eimeria* spp.—39.06%, *B. coli*—62.19%, *A. suum*—34.06%, *Oesophagostomum* spp.—27.19%, *S. ransomi*—1.56% and *Cryptosporidium* spp.—9.38%. The study revealed statistically significant (*p* < 0.05) differences between age groups, seasons, and farms for all diagnosed parasites. Further research is required to better understand the epidemiology of these infections in swine from Transylvania.

## 1. Introduction

The cost-effectiveness of raising pigs primarily depends on the health of the farmed animals. Swine diseases pose a significant economic problem throughout the world, with losses from parasitic diseases being substantial compared to those caused by bacterial and viral infections [1]. Parasites precede bacterial and viral diseases, exacerbated by the deteriorating condition of pigs [2]. Parasitic infections cause significant economic losses on swine farms by decreased production and reproduction, and also by augmented morbidity and mortality [3]. Intestinal malabsorption, impaired fertility, delayed or incomplete immunity subsequent to vaccinations, negative effects on the meat quality are all consequences such conditions can cause [4]. Pigs may subclinically harbor numerous intestinal parasites, most commonly protozoa (*Balantidium coli*, *Entamoeba* spp., *Cryptosporidium* spp.) and nematodes (*Ascaris suum*, *Trichuris suis*) [5].

The vast majority of swine in Romania, are raised on low input farms, the number of which has been registered as increasing in the last decades [6]. Organic farming depends on the ecological factors focusing on environment protection, plant health, animal health, food safety, and consumer health [7]. The free-range raising system is a type of farming where the animals, for at least part of the day, can roam freely outdoors rather than being confined in an enclosure for 24 h each day. On most farms, the outdoor areas are fenced, thus creating an enclosure. However, free-range systems usually offer the animals the opportunity for extensive locomotion and sunlight, prevented by the indoor housing systems [8].

Swine infections with gastrointestinal parasites are widely reported worldwide and are influenced by the type of swine management practices [9]. The raising of free-range pigs is common in rural areas of numerous developing countries despite its shortcomings such as poor food conversion, high mortality rates, and inferior products [10,11]. Moreover, the domestic pig is an important epizootic reservoir of parasites, exposing other animals and humans to health risks [2]. Most of the time, the course of such parasitic infections is subclinical, but symptomatic infections may occur, particularly in younger pigs [12]. The most frequent mistakes made by pig owners for parasitic infection control include the lack of fecal sample testing of animals in order to reveal particular parasite problems on the farm, the improper administration of anti-parasitic drugs, and ineffective disinfection of the premises [13].

Therefore, the identification of parasitic profiles typical to different environmental conditions, studies on parasites’ pathogenicity and subsequent development of programs to prevent/limit the spread and invasiveness of parasites are essential. Parasitological analysis of pig herds depending on the farming system is also vital for preventing the infection [2]. The current study aimed at identifying the parasitic profile of swine raised on two free-range farms in Transylvania included in three age categories. Romania has a temperate-continental climate of transitional type, with four clearly defined seasons [14], therefore systematic sampling over a year would also allow the investigation of possible seasonal trends of the identified parasitic infections.

## 2. Results

The coproparasitological examination revealed co-infections with several species of parasites, respectively, *Eimeria* spp. (Figure 1), *Balantidium coli* (Figure 2), *Ascaris suum* (Figure 3), *Trichuris suis* (Figure 4), *Oesophagostomum* spp. (Figure 5), *Strongyloides ransomi* (Figure 6) and *Cryptosporidium* spp. (Figure 7). L3 belonging to *Oesophagostomum* genus developed in the cultured fecal samples containing strongylid eggs. All the oocysts developed in the cultured samples belonged to the *Eimeria* genus.

All samples were negative by sedimentation and Blagg methods. The flotation, oocysts/egg culture, and McMaster methods showed that the prevalence and the average intensity of infections varied according to age, swine category, season, and farm.

In the weaners from farm 2 (F2), only *B. coli*, *Eimeria* spp., and *Cryptosporidium* spp. were found, while on farm 1 (F1), *Oesophagostomum* spp. and *S. ransomi* were additionally identified (Table 1). The broadest spectrum of parasites on F1 was identified during the autumn season.

In the fatteners, *B. coli*, *Eimeria* spp., *A. suum*, and *T. suis* were diagnosed on both farms (Table 2). As opposed to F1 where the most prevalent parasite infections were recorded in winter, the autumn season was that of maximal parasitic infection on F2.

In sows, *B. coli*, *Eimeria* spp., *A. suum*, *Oesophagostomum* spp., and *Cryptosporidium* spp were identified; in addition, *S. ransomi* was diagnosed on F1 (Table 3), where the broadest spectrum of parasitic infections was recorded in the farm 1.

Table 1, Table 2, Table 3, Table 4 and Table 5 provide a comparison of parasitic infections on farm 1 and farm 2, depending on age categories, parasite species, season, and the statistical significance of the differences.

## 3. Discussion

The aim of this extensive study was to evaluate the prevalence and intensity of parasites in pigs raised on free-range farms in Romania. Overall, all three age groups, weaners, growers, and sows, were almost equally parasitized by coccidia and nematodes. Similar studies, emphasizing parasitic loads in swine raised on free-range farms, were conducted in several European [3,15,16,17], African [10,18], and Asian [12,19,20,21,22,23] countries and revealed results more or less comparable with the current study.

The zoonotic protozoan *B. coli* was the most frequent parasite in all categories, with the highest prevalence (72.5%) in fattening pigs, while in weaners and sows the values were somewhat lower (70.31% and 62.19%, respectively. Similar infection rates were diagnosed in Kenya (69.6% in sows, 69.2% in fatteners and 66.7% in sows) and Greece (13.5% in weaners, 54.3% in fatteners, 81.3% in sows) [10,11,12,13,15]. In other countries, the prevalences were lower, e.g., China (18.2% in sows, 38.8% in sows, 5.7% in sows), Malaysia (22%), India (6.6–48%), Bangladesh (28.6% in weaners, 52.4% in fatteners, 38.5% in sows) and Germany (0.7% in weaners) [16,19,20,21,22,23]. The differences in prevalence could be due to some extrinsic factors such as the detection procedures, sampling sizes, farm management, climate differences, geographical separation, and intrinsic ones such as the breed, immune status, or other intercurrent diseases [22].

*Eimeria* spp. was the second most prevalent genus in all age categories recording a prevalence of 63.2% in weaners, 50.9% in fatteners, and 39.0% in sows. The infection rate was similarly recorded in all age groups of pigs in Poland but revealed a lower level of prevalence of 31.4% in weaners, 7.1% in fatteners, and 17.1% in sows, respectively [3]. In South Africa, the prevalence of coccidia infection was 88% in weaners, 75 % in fatteners, and 43.8% in sows [18]. A study performed in three African countries, South Africa, Ethiopia, and Rwanda, recorded a prevalence of *Eimeria* spp. infection varying between 5.6–88% in all categories, being the most prevalent in weaners, followed by fatteners and sows [18,24,25]. *Eimeria* spp. was identified in fattening pigs in India and China at a prevalence of 5% and 6.35%, respectively [21,26]. Another study performed in the Netherlands reported the prevalence of 50% in weaners, 7.1% in fatteners, and 87.5% in sows, respectively [17]. Various chemoprophylaxis strategies and climatic conditions that influence the resistance and embryogenesis of *Eimeria* oocysts in the environment seemed to be the main causes of the prevalence variability in the mentioned studies.

*Cryptosporidium* species were also identified on both farms, being more prevalent in weaners (18.12%) than in sows (9.38%), and absent in fatteners. Pigs are the primary host for *C. suis* and *C. scrofarum*, but *C. parvum* and *C. muris* have also been identified on some farms [27]. The worldwide prevalence differs from country to country, such as in Canada (39% in weaners and 9% in sows), China (8.7–47.9%), Argentina (9% in weaners), the Czech Republic (12%), and Ireland (15% in weaners and 13.3% in sows) [28,29,30,31,32,33]. These differences are primarily due to the examination method and sampling strategies, the differences in management systems, and the age category of the examined pigs, suckling piglets and weaners being more susceptible to *Cryptosporidium* spp. infection than adults.

Regarding helminths, *T. suis*, *A. suum*, *Oesophagostomum* spp., and *Strongyloides ransomi*, species with global distribution, were detected.

*Trichuris suis* was diagnosed in the current study only in fatteners with a prevalence of 39.06%. In Europe, the prevalence of this ceccum nematode varied between 2.9 and 11.2% in fatteners in Poland and up to 14.3% in the Netherlands [2,3,17]. In African countries (South Africa, Nigeria, and Uganda), the reported prevalence was higher compared to Europe, the values ranging from 7 to 63.9% [18,34,35,36]. The lack of anthelmintic chemoprophylaxis and the existence of an abundant infection source on the pastures shared with the African wild pigs could have been the reasons for the higher prevalence recorded in African countries. *T. suis* is widespread amongst pigs being influenced by the housing system; as such, it is more specific to the free-range farms than intensive systems [8].

Ascariasis is the major digestive parasitic infection that causes the most significant economic losses worldwide in farmed pigs, mainly on free-range farms, while its prevalence differs from one country to another. *Ascaris suum* recorded a prevalence of 63.13% in fatteners, being the second most prevalent species in this age group, and 34.06% in sows, representing the third prevailing species in the present study. An older study performed in 20 Danish herds revealed the prevalence of 15.5% in fatteners and 7.4% in sows [37]. More recently, *A. suum* showed a variable prevalence of 10% and 33% on two Danish farms, respectively [38]. A subsequent study reported *A. suum* infection in all organic farms, averaging across farm and season 48, 64, 28, and 15 % in starters, finishers, dry, and lactating sows, respectively [39]. A later study in Denmark indicated a much lower prevalence of only 28% [40]. The prevalence of milk spots in slaughterhouses in Great Britain varied between 3.29–6.85% in fatteners and 2.35–6.83% in sows [41]. The average prevalence in some free-range farms in the Netherland was 42.9% in fatteners [17]. In Albania, *A. suum* was the most common digestive parasite with a prevalence of 83.68% in weaners, 78.96 % in fatteners, and ranging from 47.02 to 53.73% in sows, with 17.65% in boars [42]. Other studies were conducted in African countries such as Zimbabwe, South Africa, Nigeria, Ethiopia, and Rwanda, where the prevalence of *A. suum* infection varied between 5.08–63.9%, with a lower prevalence in sows when compared to fatteners [2,18,24,25,34,35,36,43]. In South Africa, a study performed in 2019 showed a prevalence of 63.9% in fattening pigs [18]. Following the studies conducted in China and Greece, it was concluded that the *A. suum* infection prevalence is lower on commercial farms (0.9–16.5%) compared to free-range farms [15,26].

Species belonging to the *Oesophagostomum* genus were recorded on both farms, being more prevalent in sows (27.19%) than weaners (9.38%) and absent in fatteners. Similar values were recorded in Greece (sows—5.2%), Netherlands (weaners—12.5%, sows—37.5%), Poland (3.44—20.8%), South Africa (weaners—8%, S—43.8%), and Ethiopia (6.75%) [2,15,17,18,24,44]. Higher prevalence was found in some African countries (Zimbabwe, Rwanda, Kenya, and Uganda), varying from 54.6 to 89%, and, in Sweden, of 88% [25,34,43,45].

*Strongyloides ransomi* was rarely identified, being recorded only on F1, with a prevalence of 3.75% in weaners and 1.61% in sows. Similar results were obtained in Poland and China, where the prevalence was of 1.61%, and 6.49%, respectively [2,26]. Higher prevalence rates were diagnosed in African countries, ranging from 14% in Zimbabwe to 78.3% in Kenya [27,43]. A recent study from Cameroon also indicated a high prevalence of 34.5% [46]. The widespread use of anthelmintic programs associated with high levels of hygiene and the environmental requirements of the parasite that restrict its spread may be the reason for lower incidence in developed countries [47].

The economic losses caused by parasitic infections on swine raising generate a low interest from the research community when compared to the effort required to successfully manage and reduce their impact on swine breeding systems. However, the present study allows us to accept the hypothesis that the parasitic fauna is more affluent and contains a broader diversity of species in pigs raised on free-range farms, the prevalence and intensity being higher in such pigs than in those raised in large industrial farms. The presented results reflect the effects of rearing conditions on the susceptibility of pigs to parasitic infection as well as parasite seasonality.

## 4. Materials and Methods

The samples were collected from two free-range farms Figure 1 and Figure 2, both raising Mangalița and Bazna local swine breeds, where in 2021, Figure 1 had a pig herd of 350 animals while Figure 2 had 200 animals. The farms were located in Cluj County, a hilly area defined by abundant pastures and forests; the specific temperate-continental climate of the region is characterized by an average annual temperature between 7.5 to less than 10.0 °C and effective precipitation ranging from less than 500–600 mm/yr [48]. Drinking water for the animals was provided from a local fresh water source. The shelters were periodically cleaned throughout the year. The animals had access to outdoor areas at all times.

Three age groups were defined in each of the studied pig herds, namely weaners aged between 8 and 12 weeks, fatteners between 4 and 6 months, and sows aged from 1 to 5 years. Feces were collected from forty, randomly selected pigs from each age group during each season, this resulting in 120 samples per season and 480 feces samples during the year from one farm, respectively. A total of 960 fecal samples from the two farms were collected and examined during the experiment.

The feces samples, weighing approximately 10 g each, were collected individually from ten animals in each pen during defecation, placed in clean containers, macroscopically examined for the presence of visible parasites, then numbered and stored at 2–8 °C between 24 and 48 h, until testing.

Collected samples were examined by centrifugal sedimentation [49], flotation—Willis method, fecal smear stained by modified Ziehl-Neelsen technique [50], Blagg method, McMaster egg counting technique, and fecal cultures [51].

The morphological identification and differentiation of cultured larvae were based on several criteria such as the body size, number, shape, and arrangement of intestinal cells, type of esophagus, and length of the tail, according to the identification keys [52,53].

The individual intensity was calculated using the quantitative fecal flotation technique McMaster. In contrast, the mean intensity was determined as the arithmetic means of the number of individuals (eggs, cysts, or oocysts) of a particular parasite species per infected host in a sample.

Statistical analyses—prevalence and at least the 95% confidence interval of parasite infection were calculated for each group, season, and farm. Student’s *t*-test was used to analyze the differences among groups, seasons and farms by use of Excel program. The average and standard deviation of the mean were calculated for the number of oocysts, cysts, and eggs per gram of feces. A value of *p* ≤ 0.05 was considered statistically significant (Table 4 and Table 5).

The ontologies of pathogens and diseases are described in Table A1 (Appendix A), in accordance with the PILLOW project data management plan.

## 5. Conclusions

This study provides essential information on Transylvania’s distribution of gastrointestinal parasites in pigs. It was demonstrated that different species of gastrointestinal parasites are present in most pigs reared in free-range farms in the study area. The current information has great value to farmers, policymakers, and researchers alike, that should contribute to safer and healthier pork production for public consumption. Specifically, control strategies are needed to raise awareness among pig farmers about the negative impact of these parasites on the productivity and health of pigs and, in some cases, on human health (certain pig parasites are zoonotic).

## Figures and Tables

**Figure 1 pathogens-11-00954-f001:**
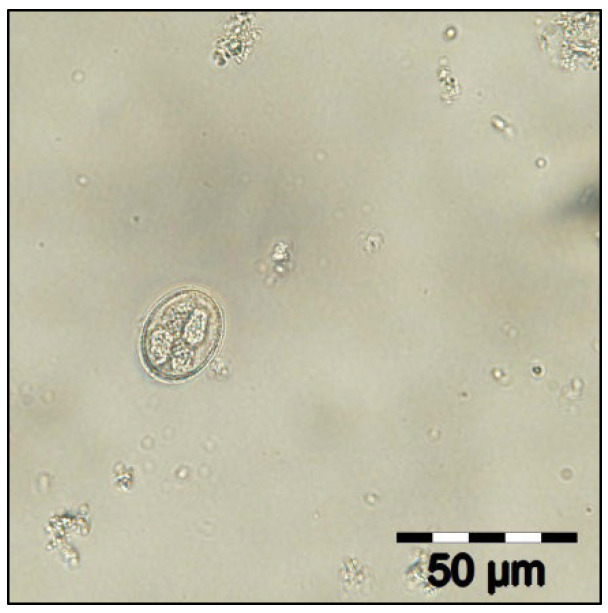
Embryonated *Eimeria* spp. oocyst.

**Figure 2 pathogens-11-00954-f002:**
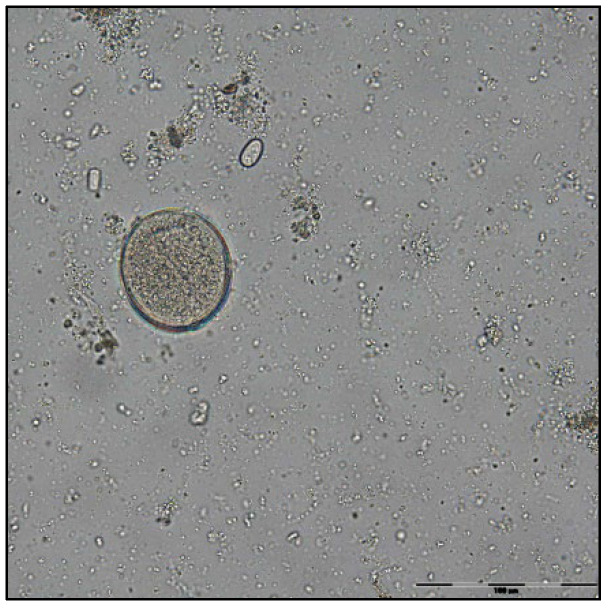
*Balantidium coli* cyst.

**Figure 3 pathogens-11-00954-f003:**
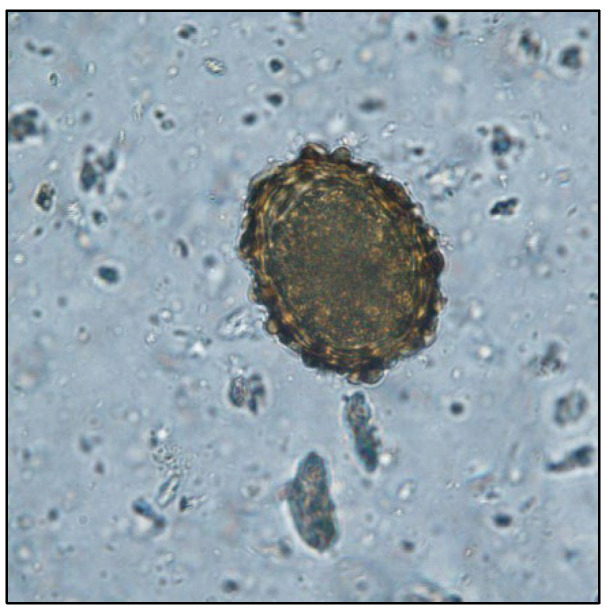
*Ascaris suum* egg.

**Figure 4 pathogens-11-00954-f004:**
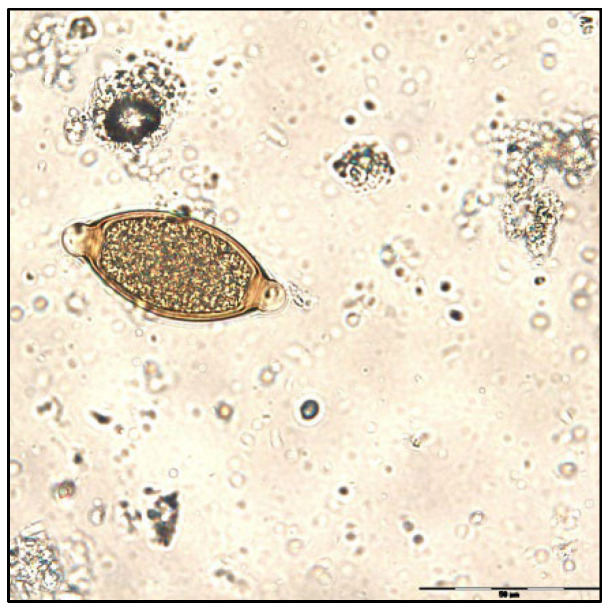
*Trichuris suis* egg.

**Figure 5 pathogens-11-00954-f005:**
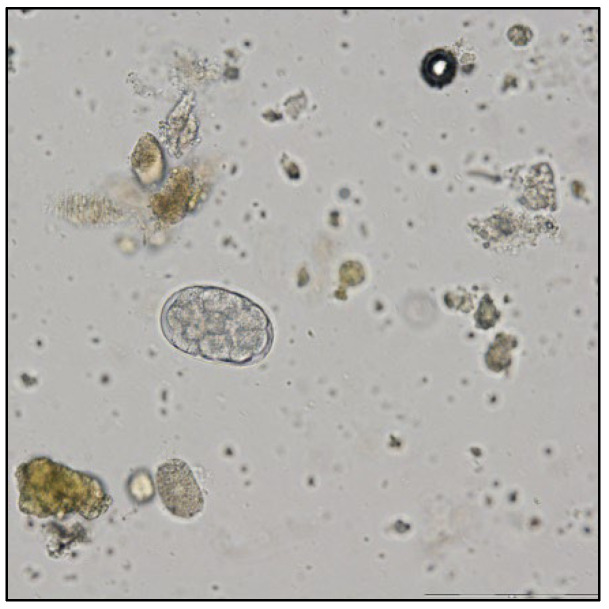
*Oesophagostomum* spp. egg.

**Figure 6 pathogens-11-00954-f006:**
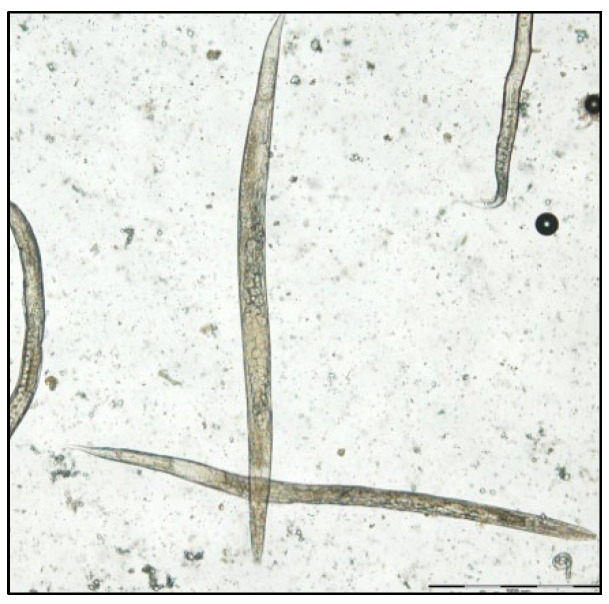
Strongyloides ransomi females.

**Figure 7 pathogens-11-00954-f007:**
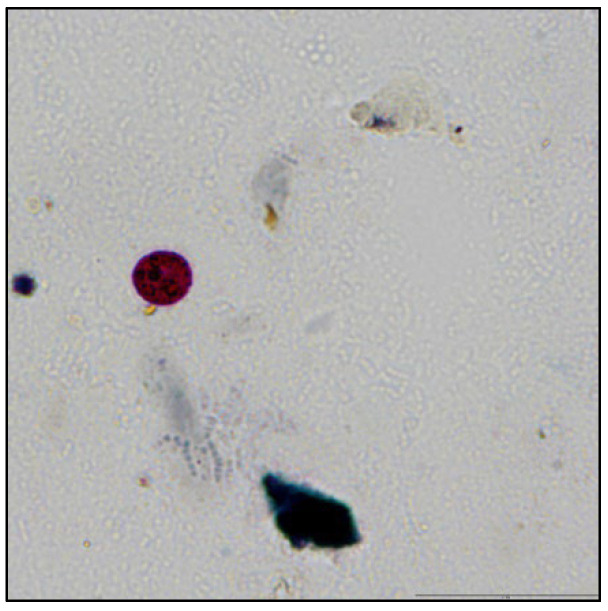
*Cryptosporidium* spp. oocyst.

**Table 1 pathogens-11-00954-t001:** The frequency (F), prevalence (P), mean intensity (MI) and standard deviation of the mean (SD) in weaners; n = number of samples.

F1												
	Spring (n = 40)			Summer (n = 40)			Autumn (n = 40)			Winter (n = 40)		
**Parasite**	**F**	**P%**	**MI (±SD)**	**F**	**P%**	**MI (±SD)**	**F**	**P%**	**MI (±SD)**	**F**	**P%**	**MI (±SD)**
*Eimeria* spp.	18	45.0	155 (±158)	32	80.0	1016 (±730)	10	25.0	300 (±351)	31	77.5	762 (±439)
*B. coli*	29	72.5	1367 (±692)	30	75.0	427 (±371)	26	65.0	723 (±638)	38	95.0	1151 (±590)
*Oesophagostomum* spp.	7	17.5	114 (±80)	4	10.0	88 (±77)	6	12.5	380 (±480)	14	35.0	457 (±199)
*S. ransomi*	-	-	-	-	-	-	12	30.0	817 (±501)	-	-	-
*Cryptosporidium* spp.	9	22.5	-	10	25.0	-	4	10.0	-	8	20.0	-
**F2**												
*Eimeria* spp.	22	55.0	1261 (±808)	23	57.5	9883 (±12563)	39	97.5	9974 (±43366)	27	67.5	2252 (±2184)
*B. coli*	32	80.0	1575 (±893)	28	70.0	1114 (±624)	23	57.5	1220 (±702)	19	47.5	963 (±603)
*Cryptosporidium* spp.	6	15.0	-	8	20.0	-	7	17.5	-	6	15.0	-

**Table 2 pathogens-11-00954-t002:** The frequency (F) prevalence (P), mean intensity (MI) and standard deviation of the mean (SD) in fatteners; n = number of samples.

F1												
	Spring (n = 40)			Summer (n = 40)			Autumn (n = 40)			Winter (n = 40)		
Parasite	**F**	**P%**	**MI (±SD)**	**F**	**P%**	**MI (±SD)**	**F**	**P%**	**MI (±SD)**	**F**	**P%**	**MI (±SD)**
*Eimeria* spp.	15	37.5	137 (±126)	16	40.0	356 (±298)	8	20.0	156 (±192)	14	35.0	354 (±251)
*B. coli*	38	95.0	1126 (±576)	18	45.0	261 (±215)	30	77.5	848 (±629)	38	95.0	1258 (±443)
*A. suum*	13	32.5	881 (±423)	12	30.0	458 (±372)	16	40.0	2681 (±3963)	34	85.0	1265 (±453)
*T. suis*	11	27.5	209 (±111)	16	40.0	181 (±129)	10	25.0	130 (±118)	21	52.5	379 (±245)
**F2**												
*Eimeria* spp.	24	60.0	679 (±524)	16	40.0	278 (±255)	40	100	4075 (±4079)	30	75.0	2693 (±2220)
*B. coli*	33	82.5	948 (±669)	30	77.5	853 (±453)	23	57.5	1008 (±758)	20	50.0	850 (±676)
*A. suum*	35	87.5	2890 (±2382)	30	77.5	2494 (±1645)	37	92.5	3973 (±2337)	24	60.0	3475 (±1964)
*T. suis*	18	45.0	847 (±580)	6	15.0	208 (±111)	24	60.0	788 (±596)	19	47.5	574 (±447)

**Table 3 pathogens-11-00954-t003:** The frequency (F), prevalence (P), mean intensity (MI) and standard deviation of the mean (SD) in sows; n = number of samples.

F1												
	Spring (n = 40)			Summer (n = 40)			Autumn (n = 40)			Winter (n = 40)		
	**F**	**P%**	**MI (±SD)**	**F**	**P%**	**MI (±SD)**	**F**	**P%**	**MI (±SD)**	**F**	**P%**	**MI (±SD)**
*Eimeria* spp.	13	32.5	127 (±133)	12	30.0	88 (±68)	8	20.0	1050 (±1165)	16	40.0	404 (±200)
*B. coli*	24	60.0	1108 (±534)	24	60.0	244 (±270)	30	75.0	823 (±619)	38	95.0	859 (±508)
*A. suum*	4	10.0	88 (±75)	12	30.0	100 (±74)	12	30.0	683 (±744)	16	40.0	472 (±310)
*Oesophagostomum* spp.	4	10.0	125 (±87)	11	27.5	182 (±129)	12	30.0	171 (±329)	15	37.5	187 (±109)
*S. ransomi*	-	-	-	3	7.5	467 (±115)	2	5.0	125 (±106)	-	-	-
*Cryptosporidium* spp	4	10.0	-	6	15.0	-	2	5.0	-	6	15.0	-
**F2**												
*Eimeria* spp.	26	65.0	5900 (±4287)	15	37.5	282 (±305)	19	47.5	560 (±365)	16	40.0	516 (±381)
*B. coli*	23	57.5	985 (±703)	9	22.5	782 (±430)	30	72.5	767 (±611)	22	55.0	857 (±602)
*A. suum*	16	40.0	1013 (±492)	18	45.0	469 (±382)	17	42.5	741 (±468)	14	35.0	643 (±361)
*Oesophagostomum* spp.	4	10.0	125 (±87)	13	32.5	219 (±180)	17	42.5	874 (±479)	11	27.5	623 (±376)
*Cryptosporidium* spp.	1	2.5	-	4	10.0	-	4	10.0	-	3	7.5	-

**Table 4 pathogens-11-00954-t004:** The statistical significance of the differences in parasitic load by season and farm (*p* value); F = farm, Sp = spring, Su = summer, A = autumn, W = winter; ssv = statistically significant value (*p* < 0.05).

** *Eimeria* ** **spp.**						
	**F1**			**F2**		
**Seasons**	**Weaners**	**Fatteners**	**Sows**	**Weaners**	**Fatteners**	**Sows**
Sp/Su	ssv	ssv	0.54	0.42	0.06	ssv
Sp/A	0.21	0.98	ssv	ssv	ssv	ssv
Sp/W	ssv	ssv	ssv	ssv	ssv	ssv
Su/A	ssv	ssv	ssv	0.53	ssv	ssv
Su/W	0.40	0.84	ssv	0.71	ssv	ssv
A/W	ssv	ssv	ssv	0.33	0.28	0.69
Sp/Sp	ssv	ssv	ssv	ssv	ssv	ssv
Su/Su	0.22	ssv	ssv	0.22	0.27	ssv
A/A	ssv	ssv	0.21	ssv	ssv	0.21
W/W	ssv	ssv	0.71	ssv	ssv	0.71
** *B. coli* **						
	**F1**			**F2**		
**Seasons**	**Weaners**	**Fatteners**	**Sows**	**Weaners**	**Fatteners**	**Sows**
Sp/Su	ssv	ssv	ssv	ssv	0.94	0.50
Sp/A	ssv	ssv	ssv	0.09	0.70	0.17
Sp/W	0.53	0.09	0.06	ssv	0.34	0.63
Su/A	0.32	ssv	ssv	0.92	0.64	0.44
Su/W	ssv	ssv	ssv	0.36	0.28	0.85
A/W	ssv	ssv	0.48	0.53	0.65	0.36
Sp/Sp	0.29	0.08	0.20	0.29	0.08	0.20
Su/Su	ssv	ssv	ssv	ssv	ssv	ssv
A/A	ssv	0.64	0.91	ssv	0.64	0.91
W/W	0.30	ssv	0.87	0.30	ssv	0.87
	** *A. suum* **				** *T. suis* **	
	**F1**		**F2**		**F1**	**F2**
**Seasons**	**Fatteners**	**Sows**	**Fatteners**	**Sows**	**Fatteners**	**Fatteners**
Sp/Su	ssv	0.77	0.83	ssv	0.40	ssv
Sp/A	0.70	0.10	ssv	0.12	0.07	0.78
Sp/W	ssv	ssv	ssv	ssv	0.12	0.27
Su/A	ssv	ssv	ssv	ssv	0.28	ssv
Su/W	ssv	ssv	ssv	0.07	ssv	0.08
A/W	0.36	0.75	0.44	0.66	ssv	0.37
Sp/Sp	0.07	ssv	0.07	ssv	ssv	ssv
Su/Su	ssv	ssv	ssv	ssv	0.48	0.48
A/A	ssv	0.09	ssv	0.09	ssv	ssv
W/W	ssv	0.14	ssv	0.14	-	0.27
	** *Oesophagostomum* ** **spp.**				** *S. ransomi* **	
	**F1**		**F2**		**F1**	
**Seasons**	**Weaners**	**Sows**	**Sows**		**Sows**	
Sp/Su	0.60	0.18	0.46		-	
Sp/A	0.43	0.79	ssv		-	
Sp/W	ssv	0.08	ssv		-	
Su/A	0.38	0.20	ssv		0.07	
Su/W	ssv	0.75	ssv		-	
A/W	ssv	0.08	0.31		-	
Sp/Sp	-	0.54	0.54		-	
Su/Su	-	0.81	0.81		-	
A/A	-	ssv	ssv		-	
W/W	-	ssv	ssv		-	

**Table 5 pathogens-11-00954-t005:** The overall prevalence (%) and frequency (F) according to age category; n = number of samples.

Parasite	Weaners (n = 320)		Fatteners (n = 320)		Sows (n = 320)	
	F	%	F	%	F	%
*Eimeria* spp.	203	63.4	163	50.9	125	39.0
*B. coli*	225	70.3	232	72.5	199	62.1
*A. suum*	-	-	202	63.1	109	34.0
*T. suis*	-	-	125	39.0	-	-
*Oesophagostomum* spp.	30	9.3	-	-	87	27.1
*S. ransomi*	12	3.7	-	-	5	1.5
*Cryptosporidium* spp.	58	18.1	-	-	30	9.3

## Data Availability

Not applicable.

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
