# Peer review of "Prevalence of Swine Gastrointestinal Parasites in Two Free-Range Farms from Nord-West Region of Romania"

_pathogens, 2022, doi:10.3390/pathogens11090954_

Round 1
Reviewer 1 Report
The article, "Prevalence of swine gastrointestinal parasites in two free-range farms from Nord-West region of Romania" reports and describes the findings of parasite profiles and percentage rates after examining 960 fecal samples from pigs raised on two different free-range farms and using various copro - detection methods. The authors then, compare rates according to age and season.
Although many such articles are currently available from different regions in this country and others, the article has considerable regional importance and should be considered for publication after major revision.
Major points
A scientifically orientated English speaking professional needs to read through the draft and correct grammatical errors.
The "Discussion" section needs to be better organized in order to describe the findings of this article compared to reports of others without losing the final targeted message to the reader. It may be better not to delve to deeply into different inter-observational findings of all reports but only reference those of major regional importance unless the differences in publication findings suggest specific environmental or herd management differences that can then be discussed.
Minor points
Abstract:
Lines 17-18: "In the last decade, it is evident that the number of free-range swine farms in Romania has increased"
Line 19: "raised on two free-range"
Line 21: What is "active sedimentation"?
Line 22: "and oocyst/egg cultures"
Line 24 and elsewhere through text: "Oesophagostomum spp"
Line 25: "Cryptosporidium spp"
Line 29: Remove parentheses after the %.
Line 34, keywords: Place in alphabetical order and change digestive parasites to "gastrointestinal parasites".
Line 38: Please cite reference for the statement.
Lines 56-57: Change to "The raising of free-range pigs is common in rural areas of many developing countries despite shortcomings like ….."
Lines 56-58: The zoonotic importance of swine diseases is repeated and should be omitted in one of the two sentences.
Lines 61-63: Again, the authors repeat that parasitic diseases reduce production in pigs (already mentioned earlier in the Introduction. Try target the importance of parasitism in fewer sentences instead of continually repeating the same idea.
Line 63: "The most frequent mistakes made by pig owners for parasitic infection control include the lack of fecal sample testing of animals in order to be aware of a particular parasite problem on the farm, the improper administration of antiparasitic drugs, and ineffective disinfection of premises [12]".
Line 67: "the profile of parasitism."
Line 72:" Systematic sampling….."
Lines 78-79: L3 belonging to Oesophagostomum genus developed in the cultured fecal samples containing strongylid eggs."
Line 81: "All samples were negative by sedimentation and Blagg methods"
The figures, although illustrative for the uninformed individual, are not necessary and do not add much to the manuscript. I would suggest removing them. If the figures are kept, species in the legends are incorrectly labeled, eg. "Cryptosporidium spp, Eimeria spp, Oesophagostomum spp."
Tables
The pdf versions of Tables 1, 2, 3 are illegible in the current form. Please resend with line numbers properly aligned. In addition, species names within all the tables are sometimes incorrectly spelled. Use one system to record probability figures. Either decide on rounding figures to the nearest decimal point or in a"x10 #n". Do not mix the forms in the same table. Sometimes probabilities are listed as, for example, "63.2%" and sometimes as, for example "9.38%". Thus, it is suggested to use two decimal points for all probabilities regardless of number, for example, "63.20 %" to continue the same pattern. In Table 5, put a full stop after "spp" in Cryptosporidium spp"
Discussion
Line 155-157: The first line needs to cite references if such a general statement is made.
Lines 158 and further on in the discussion, it is recommended that abbreviations like "W", "S", "F" are replaced by the full names of each abbreviation as it causes confusion for the reader. For example, "W" made stand for "winter" or "weaner."
Line 165" What are "other natural or artificial factors?"
Line 167 and elsewhere: Again, it is not correct to use abbreviations and additionally, it is important to add percentage after each quotation of a percentage and not only sometimes. Thus, "prevalence of 63.2% in W, 50.93% in F, and 39.06% in S"
Line 167: " The infection rate….."
Line 171: " Ethiopia, and Rwanda, recorded a prevalence of Eimeria spp. infection"
Line 173 (and possibly elsewhere in manuscript): Italicize genus or species names," Eimeria spp."
Line 173-174: The sentence is not understood.
Lines 176-177: "Various chemoprophylaxis strategies and climatic conditions…."
Line 186: Change " can" to "may".
Line 192 , 194: Change to " Danish"
Line 193-194: A. suum always produces white-spots in pigs and really this point is not necessary to mention in the statement but rather the second part of the sentence of its prevalence in these herds.
Lines 228-230: Could higher average temperatures and humidity also explain the differences? Strongyloides spp. are notoriously more widespread in the tropics.
Lines 233-237: This final section needs to be rewritten as it is not clear what the authors are trying to state.
Line 239 is better placed in the Background section and elaborated upon, specifically describing the period of each season, the average, maximum and minimum temperatures and precipitation for the sampling period (unless the significance of the statement is crucial to the choice for sampling regime and times. M&M are more appropriately specifically targeted to the animals, farm types, and region).
Line 241: Change "specimens" to animals or pigs"
Line 252: What was the sample size? How soon after sampling were the feces examined?
Line 267: How does the study provide evidence of parasite spread?
Lines 269-271: "The current information has great value to farmers, policymakers, and researchers alike, that should contribute to safer and healthier pork production for public consumption".
Line 271: "Specifically, control strategies are needed….."
Author Response
Dear reviewer,
Thank you for your professional observations and comments. We did all the suggested changes as follows:
Major points
A scientifically orientated English-speaking professional needs to read through the draft and correct grammatical errors.
A professional English speaker has reviewed the manuscript.
The "Discussion" section needs to be better organized in order to describe the findings of this article compared to reports of others
without losing the final targeted message to the reader. It may be better not to delve to deeply into different inter-observational
findings of all reports but only reference those of major regional importance unless the differences in publication findings suggest
specific environmental or herd management differences that can then be discussed.
The "Discussion" section has been reorganized according to your recommendation. We rearranged the order of the paragraphs, protozoan first, then nematodes, and introduced some new discussions.
Minor points
Abstract:
Lines 17-18: "In the last decade, it is evident that the number of free-range swine farms in Romania has increased"
We reworded the sentence according to your proposal.
Line 19: "raised on two free-range"
Accepted and modified!
Line 21: What is "active sedimentation"?
We referred to "active sedimentation" as centrifugal sedimentation, which is an active method, not a passive one, based on gravitation and requires a longer period to achieve it. We changed "active" with "centrifugal".
Line 22: "and oocyst/egg cultures"
Changed!
Line 24 and elsewhere through text: "Oesophagostomum spp"
It was corrected in the whole manuscript!
Line 25: "Cryptosporidium spp"
It was corrected in the whole manuscript!
Line 29: Remove parentheses after the %.
Removed!
Line 34, keywords: Place in alphabetical order and change digestive parasites to "gastrointestinal parasites".
Done!
Line 38: Please cite reference for the statement.
We reformulated and introduced a new reference.
Lines 56-57: Change to "The raising of free-range pigs is common in rural areas of many developing countries despite shortcomings like ….."
Done!
Lines 56-58: The zoonotic importance of swine diseases is repeated and should be omitted in one of the two sentences.
Done!
Lines 61-63: Again, the authors repeat that parasitic diseases reduce production in pigs (already mentioned earlier in the Introduction. Try target the importance of parasitism in fewer sentences instead of continually repeating the same idea.
We rephrased according to your comment.
Line 63: "The most frequent mistakes made by pig owners for parasitic infection control include the lack of fecal sample testing of animals in order to be aware of a particular parasite problem on the farm, the improper administration of antiparasitic drugs, and ineffective disinfection of premises [12]".
Modified!
Line 67: "the profile of parasitism."
Modified!
Line 72:" Systematic sampling….."
Modified!
Lines 78-79: L3 belonging to Oesophagostomum genus developed in the cultured fecal samples containing strongylid eggs."
Modified!
Line 81: "All samples were negative by sedimentation and Blagg methods"
Modified!
The figures, although illustrative for the uninformed individual, are not necessary and do not add much to the manuscript. I would suggest removing them. If the figures are kept, species in the legends are incorrectly labeled, eg. "Cryptosporidium spp, Eimeria spp, Oesophagostomum spp."
We have chosen to keep the figures even if the images of these parasites are well-known; we saw that other published articles by MDPI kept the pictures. Of course, we corrected the parasites' names.
Tables
We modified the tables keeping only one decimal. I also corrected the names of the species. Regarding the lines and their visibility, the Pathogens inserted them, not us, the authors. I hope that in the new form, they will be more visible and legible.
Discussion
Line 155-157: The first line needs to cite references if such a general statement is made.
References were inserted!
Lines 158 and further on in the discussion, it is recommended that abbreviations like "W", "S", "F" are replaced by the full names of each abbreviation as it causes confusion for the reader. For example, "W" made stand for "winter" or "weaner."
The abbreviations W, F, and S, were replaced throughout the manuscript with their meanings, weaners, fatteners, and sows, to avoid confusion!
Line 165" What are "other natural or artificial factors?"
We changed "natural" and "artificial" with "extrinsic" (they depend on the growing conditions) and "intrinsic" (animal-related).
Line 167 and elsewhere: Again, it is not correct to use abbreviations and additionally, it is important to add percentage after each quotation of a percentage and not only sometimes. Thus, "prevalence of 63.2% in W, 50.93% in F, and 39.06% in S"
Done!
Line 167: " The infection rate….."
Done!
Line 171: " Ethiopia, and Rwanda, recorded a prevalence of Eimeria spp. infection"
Modified!
Line 173 (and possibly elsewhere in manuscript): Italicize genus or species names," Eimeria spp."
Italicized throughout the manuscript!
Line 173-174: The sentence is not understood.
We reworded it as follows: "Eimeria spp. was identified in fattening pigs in India and China at a prevalence of 5% and 6.35%, respectively."
Lines 176-177: "Various chemoprophylaxis strategies and climatic conditions…."
Modified!
Line 186: Change " can" to "may".
Changed!
Line 192 , 194: Change to " Danish"
Changed!
Line 193-194: A. suum always produces white-spots in pigs and really this point is not necessary to mention in the statement but rather the second part of the sentence of its prevalence in these herds.
Modified as follows: "More recently, A. suum showed a variable prevalence of 10% and 33% on two Danish farms, respectively".
Lines 228-230: Could higher average temperatures and humidity also explain the differences? Strongyloides spp. are notoriously more widespread in the tropics.
Modified as follows: "The widespread use of anthelmintic programs associated with high levels of hygiene and the environmental requirements of the parasite that restrict its spread may be the reason for lower incidence in developed countries".
Lines 233-237: This final section needs to be rewritten as it is not clear what the authors are trying to state.
We have partially rewritten the final section. It was meant to be a kind of discussion's conclusion, not of the manuscript.
Line 239 is better placed in the Background section and elaborated upon, specifically describing the period of each season, the average, maximum and minimum temperatures and precipitation for the sampling period...
We entered some of the requested data at M&M. However it is virtually impossible to obtain such detailed data, restricted to a certain area.
Line 241: Change "specimens" to animals or pigs"
Changed!
Line 252: What was the sample size? How soon after sampling were the feces examined?
We introduced: "feces samples, weighing approximately 10 grams each", and "stored at 2-8℃ between 24 and 48 hours, until testing".
Line 267: How does the study provide evidence of parasite spread?
Indeed, the study provides less evidence of parasite spread. As such, we kept only: "This study provides essential information on Transylvania's distribution of gastrointestinal parasites in pigs."
Lines 269-271: "The current information has great value to farmers, policymakers, and researchers alike, that should contribute to safer and healthier pork production for public consumption".
Modified!
Line 271: "Specifically, control strategies are needed…..
Changed!
Reviewer 2 Report
Congrats for your manuscript, you are almost there.
On table 1 you present "average intensity values". Please, consider if it would be better to use the term "mean intensity" that literally mean the same but it is somehow more habitual. Likewise you should include in M&M the formula you used for the Average intensity / Mean intensity calculations, also the "probability value" you used later in other table.
The tables should include N numbers. I know they are explained later in M&M but providing you chose to leave M&M for the end, you should add "n" numbers otherwise you puzzle the reader.
Table 4 is unacceptable in its current format. It is unintelligible Please, remove meaningless decimals and fix the exponential format to something that can be understood. Does it mean something the red colour, sorry I missed the explanation. If not explained or clarified, please, use one colour. Work on the format to align the values. The tables are not curated and need a format revision. There are other tables in which meaningless decimals should be removed in other to facility the reader to extract info.
Reagrds
Author Response
Dear reviewer,
Thank you for your professional comments!
We reconsidered the term "average intensity" and changed it with "mean intensity".
We also included in the M7M section an explanation of the "mean intensity" and how it was calculated, as follows: "The individual intensity was calculated using the quantitative fecal flotation technique McMaster. In contrast, the mean intensity was determined as the arithmetic means of the number of individuals (eggs, cysts, or oocysts) of a particular parasite species per infected host in a sample.".
We changed the "probability value" with "p-value", which is a well-known parameter.
It is almost impossible to enter the value of N in all the tables. Still, we introduced in table 5 the frequency (F), which indicates the number of parasitized animals from the total of those examined.
Indeed, the format of table 4 was unacceptable. I modified it, keeping only two decimal places, and for the statistically significant values, we introduced the abbreviation ssv, meaning "statistically significant value". The other tables were also modified according to your recommendations.
Reviewer 3 Report
My recommendations and revisions on the MS are listed below.
Line 25: Criptosporidium should be corrected as Cryptosporidium.
Line 58-60: The use of the “vector” expression for vertebrate animals is incorrect. Vector is used for arthropod species, such as mosquitoes, ticks, sandflies, blackflies…
Line 24, 27, 30: Esophagostomum should be corrected as Oesophagostomum
Table 3 and Table 4: Esofagostomum should be corrected as Oesophagostomum.
Line 78-80: How the authors differentiate and separate Oesophagostomum genus from other common parasites such as Hyostrongylus rubidus in pigs. This statement should be given in more detail.
Author Response
Dear reviewer,
Thank you for your professional review.
We made the following changes:
Line 25: Criptosporidium should be corrected as Cryptosporidium.
We corrected "Criptosporidium" with "Cryptosporidium" in the whole manuscript.
Line 58-60: The use of the “vector” expression for vertebrate animals is incorrect.
You are right! We deleted the term "vector" from the respective wording!
Line 24, 27, 30: Esophagostomum should be corrected as Oesophagostomum
Table 3 and Table 4: Esofagostomum should be corrected as Oesophagostomum.
We corrected "Esofagostomum" with "Oesophagostomum" in the whole manuscript.
Line 78-80: How the authors differentiate and separate Oesophagostomum genus from other common parasites such as Hyostrongylus rubidus in pigs. This statement should be given in more detail.
We introduced in the M&M section the following phrase: "The morphological identification and differentiation of cultured larvae were based on several criteria such as the body size, number, shape, and arrangement of intestinal cells, type of esophagus, and length of the tail, according to the identification keys [53, 54]."
Round 2
Reviewer 1 Report
The article has been improved and can be published now.after the tables are chages are made to make them more legible as they are still illegible in the last copy sent. Also read again to check typos. Change Line 506: "prevalencewas"
Author Response
Dear reviewer,
Thank you again for your perseverance in helping us to improve the manuscript qualitatively. Your comments and advice were helpful for us!
"The article has been improved and can be published now after the tables changes are made to make them more legible as they are still illegible in the last copy sent. "
In this regard, we modified the tables by introducing new columns (F), meaning frequency, thus giving a better image of the prevalence and how we obtained it. We also mentioned in the head of tables the number (n) of samples for each age group and season.
To make the manuscript (and tables) more legible, we accepted all the track changes that appeared during the revision of the manuscript, thus resubmitting a final form.
"Also, read again to check typos" - checked!
"Change Line 506: "prevalencewas" - changed!